# The right amygdala and migraine: Analyzing volume reduction and its relationship with symptom severity

Shota Kosuge[1,2], Yuri Masaoka[1]*, Hideyo Kasai[2], Motoyasu Honma[1], Kouzou Murakami[4], Nobuyuki Yoshii[4], Keiko Watanabe[1,2], Takaaki Naito[4], Miku Kosuge[1,3], Misa Matsui[1], Daiki Shoji[1,2], Syunsuke Sakakura[1], Hidetomo Murakami[2], Masahiko Izumizaki[1]

1 Department of Physiology, Showa University School of Medicine, Tokyo, Japan, 2 Department of Neurology, Showa University School of Medicine, Tokyo, Japan, 3 Department of Respiratory Medicine, Showa University Fujigaoka Hospital, Yokohama, Japan, 4 Department of Radiology, Showa University School of Medicine, Tokyo, Japan

* faustus@med.showa-u.ac.jp

**Data Availability Statement:** All relevant data are within the manuscript and its Supporting Information files.

## Abstract

This study aimed to explore the relationship between gray matter volume changes and various clinical parameters in patients with migraine, focusing on symptom severity, quality of life, and states of depression and anxiety. Using a case-control design, we examined 33 patients with migraine, with or without aura, and 27 age-matched healthy subjects. We used magnetic resonance imaging to assess the volumes of 140 bilateral brain regions. Clinical evaluations included the Migraine Disability Assessment, the Migraine Specific Quality of Life Questionnaire, the Center for Epidemiologic Studies Depression scale, Spielberger's State and Trait Anxiety scales, and the Japanese version of the Montreal Cognitive Assessment. We compared the scores of these measures between migraine patients and healthy controls to examine the interplay between brain structure and clinical symptoms. Significant volumetric differences were observed in the pallidum and amygdala between migraine patients and healthy individuals. The reduction in the right amygdala volume correlated significantly with migraine severity as measured by the Migraine Disability Assessment. Path analysis revealed a model where Migraine Disability Assessment scores were influenced by Migraine Specific Quality of Life Questionnaire outcomes, which were further affected by depression, anxiety, and a low right pallidum volume. Our findings suggest that the chronicity and severity of migraine headaches specifically affect the right amygdala. Our path model suggests a complex relationship whereby migraine disability is strongly influenced by quality of life, which is, in turn, affected by psychological states, such as anxiety and depression.

**Funding:** JSPS KAKENHI Grant (Number 20K03482). The funders had no role in study design, data collection and analysis, decision to publish, or preparation of the manuscript.

**Competing interests:** The authors have declared that no competing interests exist.

## Introduction

Migraine (MG), a prevalent neurological disorder marked by severe headaches and an array of neurological symptoms, poses a significant public health challenge worldwide [1, 2]. MG significantly impairs daily activities and exacerbates depression and anxiety because of the burdens of headache-related disabilities [3]. These repercussions significantly diminish patients' quality of life, social engagements, and family interactions.

The Third Edition of the International Classification of Headache Disorders [4] states that primary headache does not typically feature reductions in brain structure. However, numerous neuroimaging studies have reported gray matter volume reductions in MG patients [5–7]. Specific areas of volume reduction in patients with MG include the insula, middle and inferior frontal cortex, and cingulate cortex [5]. Chronic MG patients, when compared with those with episodic MG, have shown focal gray matter decline in regions including the bilateral anterior cingulate cortex, left amygdala, left parietal operculum, left middle and inferior frontal gyri, right inferior frontal gyrus, and bilateral insula [6]. Furthermore, a meta-analysis by Hu et al. [7] found that the middle and inferior frontal cortices are significantly reduced in MG patients. Moreover, in the basal ganglia, the bilateral globus pallidus and left putamen are considerably smaller in MG patients with aura [8]. Imaging studies have also revealed myriad changes in brain function due to MG attacks, such as increased cortical excitability [9], elevated cerebral blood flow [10, 11], and alterations of pain modulation systems [12, 13]. The multifaceted pathophysiology of MG involves intricate interactions among cortical, subcortical, and brainstem structures.

Our aim was to elucidate the association between gray matter volume changes and clinical factors, such as symptom severity, quality of life, and emotional states of depression and anxiety. We sought to understand the relationship between gray matter volume and MG symptom severity, with particular emphasis on the potential emotional regulatory implications for MG patients.

## Materials and methods

### Participants

The study was a case-control study. Sample size was determined according to the effect size of previous studies investigating brain volumes in young and older adult subjects [14]. Thirty-three patients with MG participated in the study. Eligibility criteria were as follows: 1) aged 21 to 60 years, 2) experienced MG attacks for a minimum of 2 years, 3) no clear structural pathology or hyperintense white-matter lesions on previous magnetic resonance imaging (MRI) scans, 4) no comorbid basal ganglia disorder or history of other neurological disorders. We recruited 27 age-matched healthy controls (aged 21–60 years) from clinical staff and their family and friends, none of whom had a history of neurological or metabolic diseases. None of the control subjects had experienced any forms of headache apart from occasional tension-type headaches or had family members diagnosed with MG. To avoid bias, three neurologists evaluated all of the participants to ensure adherence to the selection criteria. Participants were recruited for the study from December 2021 to October 2023. The study was approved by the Institutional Review Board of Showa University Hospital and the Ethical Committees of Showa University School of Medicine (clinical trial identifier 3485). All participants provided written informed consent before participating. The procedures aligned with the principles of the Declaration of Helsinki.

The MG patients provided details of MG duration, MG onset, presence or absence of aura, and the manifestation of visual and somatosensory symptoms linked to MG. All participants

underwent the following assessments: the Migraine Disability Assessment (MIDAS) [15], the Migraine Specific Quality of Life Questionnaire (MSQ) [16], the Center for Epidemiologic Studies Depression scale (CESD) [17], and Spielberger's State and Trait Anxiety scales (STAI) [18]. In addition, to evaluate cognitive function, the Japanese version of the Montreal Cognitive Assessment (MoCA-J) [19] was administered.

## MRI data collection and image analysis

MRI scans were acquired at Showa University Hospital (Tokyo, Japan) using a Siemens 3 Tesla MAGNETOM Prisma fit scanner (Siemens, Erlangen, Germany) equipped with a 64-channel phased-array coil. We used the following parameters for anatomical imaging: T1-weighted three-dimensional magnetization-prepared rapid gradient-echo sequence (flip angle 9˚; repetition time 2300 ms; echo time 2.98 ms; matrix size 256 × 256).

Image analysis was conducted using the FreeSurfer (version 6.0) automated neuroanatomical analysis software (http://surfer.nmr.mgh.harvard.edu). Gray matter volumes were extracted from each T1-weighted image using the recon-all command in FreeSurfer. The software suite enabled automatic volumetric segmentation [20], cortical surface reconstruction [21–23], and parcellation [24, 25] after initial image preprocessing, which comprised motion correction, non-brain-tissue removal, intensity normalization, affine alignment to Montreal Neurological Institute coordinates, and Talairach transformation. After registration to Montreal Neurological Institute space, all boundaries were visually verified using the Freeview and TkMedit graphic tools available in FreeSurfer. A pair of experienced neurologists reviewed all volumetric partitions for precision and made minor modifications using the TkMedit editing functionality. Manual corrections were confined to eliminating non-brain tissue that appeared within the cortical boundary, according to our written guidelines established to ensure consistency. Most errors were located in the paracentral lobule and near the cerebellar tentorium. After any necessary adjustments were made, the images were reprocessed using the recon-all command. According to quality control studies of structural images [26], differences between pre- and post-quality control measurements are small. Nevertheless, we ensured this by comparing all 140 gray matter volumes from the stats files using a paired t-test (after testing Shapiro–Wilk to confirm the normality of the distribution) and a reliablility analysis between pre- and post-manual quality control. There were no differences in gray matter volumes between pre- and post-manual quality control (all $P > 0.05$) across all participants. We confimed that Cronbach's alpha was all >0.9 (all mean and standard deviation were indicated in S1-S5 Tables in S1 File). We used the automatically extracted volume to ensure reproducibility.

## Statistical analysis

All data extracted from FreeSurfer were input into SPSS Statistics (IBM SPSS Statistics, version 23.0, IBM Corp., Armonk, NY, Chicago, USA). This included the assessed all data. SPSS was used for all statistical analyses. The gray matter volumes of 140 regions (as detailed in S1–S5 Tables in S1 File) were measured using FreeSurfer. All data were first analyzed using the Shapiro–Wilk test to test the normality of the distribution.

Differences in age, years of education, and MIDAS, MSQ, and MoCA-J scores (Shapiro–Wilk, $P < 0.05$) were analyzed using the non-parametric Mann–Whitney test, and CESD and STAI scores (Shapiro–Wilk, $P > 0.05$) were analyzed using parametric t-tests.

Group comparisons for intracranial volume (ICV) were analyzed using analysis of covariance (ANCOVA), with sex and years of education as covariates. Differences in left and right gray matter volumes between groups were assessed using ANCOVA, with sex, years of education, and ICV as covariates. Because there was no difference in volume between patients with

MG with or without aura (as noted in the results section), the presence of aura was not included as a covariate for the ANCOVA.

The main objective here was to examine the relationship between symptom severity and the neural substrates underlying symptom severity. Thus, we constrained our hypotheses according to the anatomical connectivity relevant to the MIDAS. As such, gray matter volume differences between MG patients and controls were analyzed using ANCOVA without multiple comparison correction (e.g., family-wise error correction) to identify brain regions relevant to MG. Volumes were compared between MG patients with (N = 11) and without aura (N = 22) in a similar manner. To understand the relationship between symptom severity, as measured by the MIDAS, and reduction in brain volume, we applied forced entry multiple regression and examined the association between MIDAS score and the volumes of the bilateral pallidum and amygdala. MIDAS score was treated as the dependent variable.

Before proceeding with the path analysis, partial correlation tests were conducted within groups to refine our hypotheses. Partial correlation analyses, accounting for ICV and sex, were performed between the variables of the MIDAS, MSQ, CESD, state anxiety, and trait anxiety scores, years of education, and bilateral pallidum and amygdala volumes.

Path analysis was performed to examine the interactions between the variables of MIDAS, MSQ, CESD, state anxiety, and trait anxiety scores and brain areas with reduced volume. We used Amos version 27.0 (IBM Corp.) for the path analysis. Statistical significance was set at an adjusted $P < 0.05$. Path analysis estimates the magnitude and significance of the presumed causal links between variable sets. One benefit of path analysis is its ability to gauge indirect effects. The structural equation modeling software IBM SPSS Statistics Amos (version 23.0, IBM Corp.) determined the statistical significance of the path coefficients. Straight arrows linking the variables indicate the direction of causal relationships between variables. Double-headed curved arrows indicate correlations between exogenous variables. Standardized beta coefficients are indicated by straight arrows.

Before drawing conclusions from the path analysis results, we calculated several indices to evaluate the validity of the results: the goodness of fit index, root mean square error of approximation, comparative fit index, and Bollen–Stine bootstrap. A Bollen–Stine bootstrap $P$-value $> 0.05$ was considered appropriate for the model.

### Data availability

Individual participant data and a data dictionary will be made available, subject to a data-sharing agreement, for further prespecified analyses.

### Results

Demographic data are presented in Table 1.

No significant group differences were observed for age (z = −0.2, $P = 0.81$) or MoCA-J score (z = −1.7, $P = 0.18$). However, the MG group had significantly higher MIDAS (z = −6.1, $P < 0.001$), MSQ (z = −6.1, $P < 0.001$), CESD (t = −4.1, $P < 0.001$), state anxiety (t = −3.9, $P < 0.001$), and trait anxiety (t = −3.6, $P < 0.01$) scores than controls. There were no significant differences in any variables between MG patients with and without aura (all $P > 0.05$). There was no significant difference in age (z = −2.2, $P = 0.81$) but there was a significant difference in years of education (z = −2.6, $P = 0.05$).

Initially, the brain volumes of all 140 gray matter regions and ICVs were compared between MG patients and controls. The homogeneity of variance was equal across groups as confirmed by Levene's test (all $P > 0.05$) in the ANCOVA, which indicated that parametric analysis was suitable. ICV differed significantly between the two groups ($F_{1,54} = 1.58$, $P = 0.21$, $\eta p^2 = 0.02$).

**Table 1. Demographic data.**

|  | MG patients (15–64) | Healthy Subjects (19–60) |
|---|---|---|
| Total(M/F), No. | 33 (Female, 29/Male, 4) | 27(Female, 23/Male, 4) |
| Age, y | 38.9 ± 14.4 | 37.8 ± 13.9 |
| Handedness(R/L/Ambi) | (Right, 30/Left, 2/Ambidextrous, 1) | (Right, 23/Left, 4) |
| Years of education | 14.6 ± 2.7 ** | 17.4 ± 3.6 |
| MIDAS | 27.1 ± 35.7 *** | 0.6 ± 2.2 |
| With aura | 11/33 | 0 |
| MSQ | 35.4 ± 15.6 *** | 14.8 ± 4.9 |
| CESD | 15.2 ± 9.7 ** | 6.8 ± 4.9 |
| STAI state | 45.3 ± 12.3 ** | 34.2 ± 8.3 |
| STAI trait | 46.8 ± 11.7 ** | 36.9 ± 8.2 |
| MoCA-J | 27.6 ± 1.3 | 28.2 ± 1.5 |

MG, migraine; M, male; F, female; MIDAS, Migraine Disability Assessment; MSQ, Migraine Specific Quality of Life Questionnaire; CESD, Center for Epidemiologic Studies Depression scale; STAI, Spielberger's State and Trait Anxiety scale; MoCA-J, Japanese version of the Montreal Cognitive Assessment.

** $P < 0.001$

*** $P < 0.0001$

Of the 140 regions, the bilateral pallidum (left: $F_{1,53} = 5.76$, $P = 0.02$, $\eta p^2 = 0.09$; right: $F_{1,53} = 6.42$, $P = 0.01$, $\eta p^2 = 0.11$) and amygdala (left: $F_{1,53} = 18.58$, $P < 0.001$, $\eta p^2 = 0.26$; right: $F_{1,53} = 10.77$, $P = 0.002$, $\eta p^2 = 0.16$) were significantly smaller in the MG group than in the control group. The mean volumes and standard deviations of these regions are shown in Table 2.

Those of all other regions are provided in S1-S5 Tables in S1 File.

We also compared brain volumes between MG patients with and without aura. No significant differences were observed for ICV, bilateral pallidum volume (left: $F_{1,26} = 0.17$, $P = 0.67$, $\eta p^2 = 0.007$; right: $F_{1,26} = 0.001$, $P = 0.99$, $\eta p^2 = 0.001$), or bilateral amygdala volume (left: $F_{1,26} = 0.29$, $P = 0.09$, $\eta p^2 = 0.1$; right: $F_{1,26} = 0.06$, $P = 0.79$, $\eta p^2 = 0.003$) between the two MG subgroups.

For the multiple regression analysis, we examined multicollinearity for each independent variable to ensure the variables were not correlated. The variance inflation factor was < 5 for all independent variables (left pallidum: 3.2, right pallidum: 2.9, and bilateral amygdala: 1.6).

Multiple regression analysis revealed that the volume of the right amygdala was significantly associated with the MIDAS score ($\beta = -0.56$, $t = -2.45$, $P = 0.02$; Fig 1 and Table 3).

In the exploratory partial correlation analysis, the MIDAS score was negatively correlated with the volume of the right amygdala ($r = -0.4$, $P = 0.04$), and MIDAS score was positively

**Table 2. Significant volume reductions in the bilateral pallidum and amygdala in migraine patients.**

|  | MG patients | Healthy Subjects |
|---|---|---|
| Intracranial volume | 1469765 ± 107757 | 1432214 ± 114964 |
| L. Pallidun | 1916 ± 183* | 2026 ± 179 |
| R. Pallidum | 1887 ± 156* | 1965 ± 166 |
| L.Amygdala | 1563 ± 135** | 1672 ± 197 |
| R. Amygdala | 1700 ± 149** | 1795 ± 216 |

L, left; R, right; MG, migraine

* $P < 0.05$

** $P < 0.001$

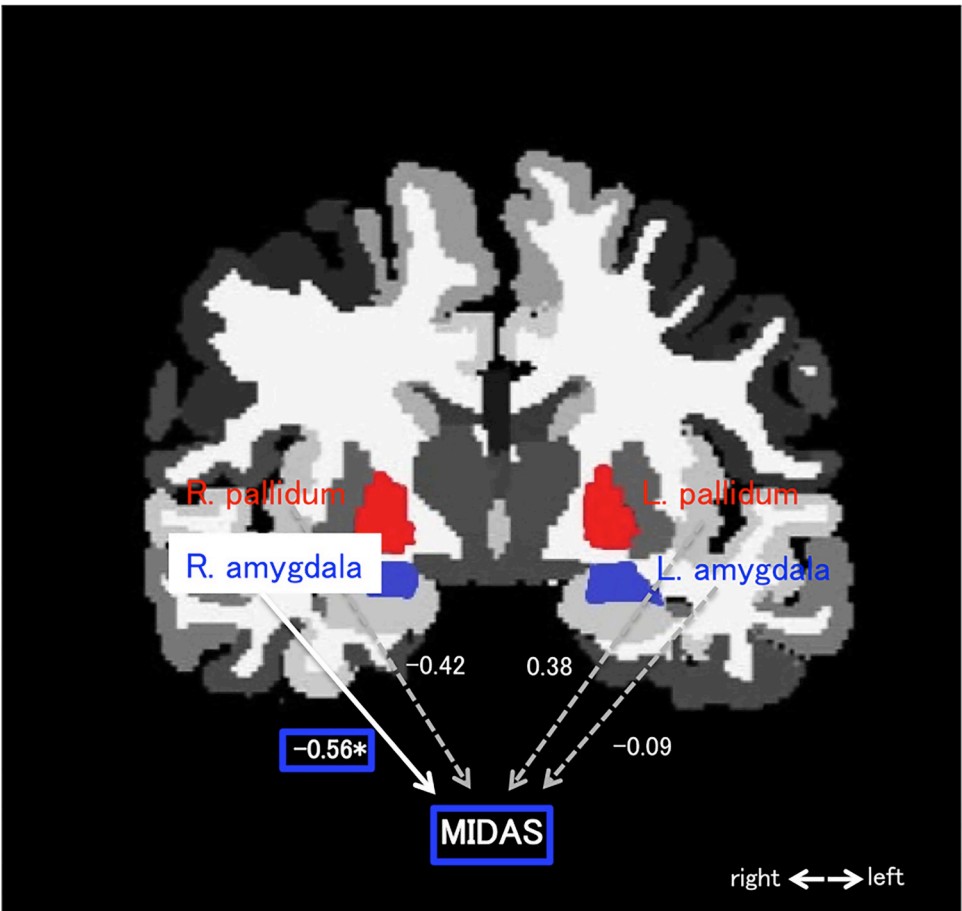

**Fig 1.** The pallidum (red) and amygdala (blue) are displayed on a coronal section extracted from the FreeSurfer viewer, superimposed with the multiple regression results. A forced-entry multiple regression was used to examine associations between the Migraine Disability Assessment (MIDAS) score and the left and right pallidum and amygdala volumes. The MIDAS score was designated as the dependent variable. A significant association was found between the volume of the right amygdala and the MIDAS score, which suggested that a lower amygdala volume correlated with a higher MIDAS score.

correlated with the MSQ score (r = 0.6, P = 0.001). Additionally, the MSQ score was positively correlated with the CESD score (r = 0.36, P = 0.05), which was correlated with both state (r = 0.82, P < 0.001) and trait anxiety scores (r = 0.74, P < 0.0001; S6 Table in S2 File). However, these correlations did not survive false discovery rate multiple comparison correction at P < 0.05. No significant correlations were observed in the control group (S7 Table in S2 File).

**Table 3. Multiple regireesion results.**

| Model | | Standardized Coefficients | t | *P* value |
|---|---|---|---|---|
| | | Beta | | |
| | (Constant) | | | |
| | Left-Pallidum | 0.386 | 1.201 | 0.242 |
| | Right-Pallidum | -0.421 | -1.377 | 0.182 |
| | Left-Amygdala | 0.096 | 0.416 | 0.681 |
| | Right-Amygdala | -0.561 | -2.427 | 0.023 |

For the path analysis, we initially tested the full model, which included the bilateral pallidum and amygdala volumes and the MIDAS, MSQ, CESD, state anxiety, and trait anxiety scores. The final model was refined by removing the non-significant paths (goodness of fit index = 0.9, root mean square error of approximation = 0.16, comparative fit index = 0.91, Bollen–Stine bootstrap: $P$ = 0.12).

In the path model (Fig 2), the three exogenous variables (i.e., state anxiety score, CESD score, and right pallidum volume) were modeled as being correlated, as depicted by the

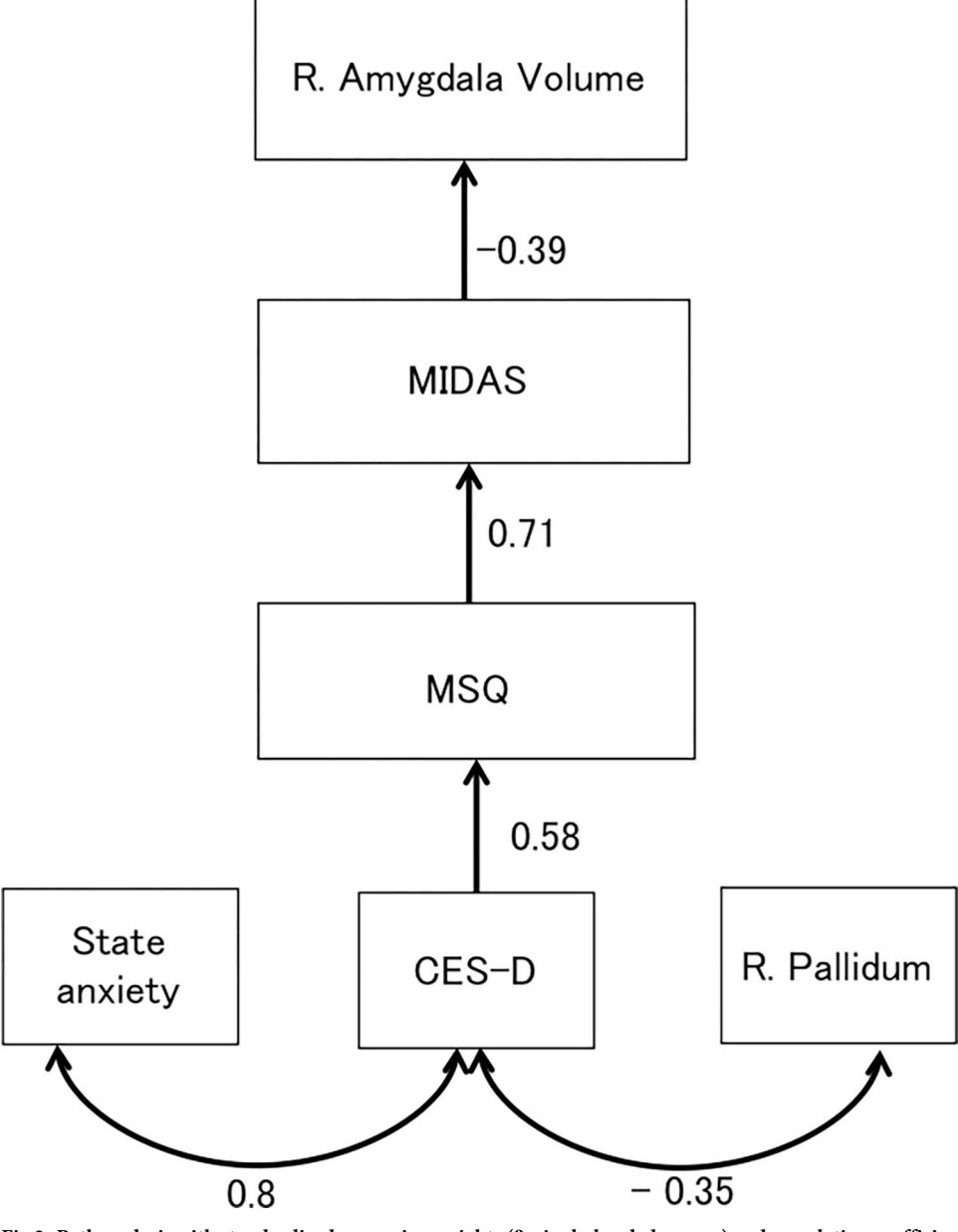

**Fig 2. Path analysis with standardized regression weights (β; single-headed arrows) and correlation coefficients (double-headed arrows) for the migraine (MG) group.** The Migraine Disability Assessment (MIDAS) score followed a standardized direct path to the right amygdala, which suggested that a higher MIDAS score had an impact on the right amygdala volume. The MIDAS score was directly influenced by the Migraine Specific Quality of Life Questionnaire (MSQ) score, and the MSQ score, in turn, had a direct path originating from the Epidemiologic Studies Depression Scale (CESD) score. The CESD score was correlated with the state anxiety score and the right pallidum volume.

double-headed arrows (indicated r as a correlation coefficient). The endogenous dependent or independent variables were connected by one single-headed arrow (indicated as betas [standardized regression weights]).

There was a covariate relationship between the CESD score and state anxiety score (r = 0.81, $P < 0.001$) and the right pallidum volume (r = −0.35, $P = 0.01$). The CESD score had a direct effect on the MSQ score (β = 0.58, $P < 0.001$). The MSQ score had an impact on the MIDAS score (β = 0.71, $P < 0.001$), and the MIDAS score had an impact on the right amygdala (β = −0.38, $P = 0.002$). This suggested that a higher MIDAS score corresponded with a smaller right amygdala volume. The standardized total and indirect effect values are provided in Table 3. The full results, including the total and indirect effects, are provided in S8 Table in S3 File.

## Discussion

Our results revealed important structural differences in the pallidum and amygdala between MG patients and healthy controls. Specifically, we identified a right amygdala volume reduction in MG patients, which impacted MG severity, as measured by the MIDAS. This relationship between the right amygdala volume and the MIDAS score suggested that higher MG severity corresponded to a smaller right amygdala volume. This finding raises several questions regarding the underlying mechanisms and implications. In terms of the direction of causality, it remains uncertain whether the lower right amygdala volume stems from the intensity of MG symptoms or whether a smaller amygdala volume exacerbates MG severity. Results of our stepwise path analysis, in which non-significant paths were excluded and the directions of paths among different factors were adjusted, suggested that lower amygdala volume is influenced by MG severity. One interpretation is that the reduction in right amygdala volume arises from chronic MG episodes, coupled with the associated stress and pain. The amygdala plays a key role in the processing of anxiety and stress [27] and is an important node of the pain network [6]. Thus, chronic activation of the amygdala may instigate alterations over time. Furthermore, reduced amygdala volumes may be associated with alterations in emotional regulation and stress responses, which may heighten susceptibility to intense headaches and MGs.

In the comparative group analysis, we identified bilateral volume reductions in both the amygdala and pallidum. Additionally, we found associations between the right amygdala and the MIDAS score and between the right pallidum and the CESD score. Lateralization of the amygdala in pain has been reported previously [28], and it has been suggested that the amygdala's role in pain originated from the right hemisphere. Indeed, Carrasquillo and Gereau [29] posited that, although nociceptive inputs predominantly influence both amygdalae, the right amygdala primarily processes nociceptive and emotional pain components. The amygdala activates continuously in response to pain, with a right-sided predominance for the regulation of pain hypersensitivity [30]. In addition, Jassar et al. [31, 35] found a correlation between diminished endogenous μ-opioids in the right amygdala and headache severity during a cutaneous thermal pain challenge [33]. In line with these reports, the pain sensation of MGs in our patients may have had a greater impact on the right amygdala than the left. However, the mechanism underlying the rightward reduction in amygdala volume in MG patients remains uncertain and requires further research.

### Assumptions from the path model

The path analysis also revealed several mediating factors that connected MG severity to lower amygdala volume. Our findings suggested that the MIDAS score, which represents MG

severity, is shaped by patients' quality of life, as measured by the MSQ. The MSQ is, in turn, influenced by exogenous factors, such as depression, anxiety, and the right pallidum volume.

The association between the right pallidum volume and depression in MG patients was an intriguing result. This suggested that those with elevated depression levels tended to have lower pallidum volumes. The pallidum is a structure within the basal ganglia that plays a pivotal role in motor control and cognitive and emotional functions [32]. Depressed patients have been shown to exhibit structural and functional anomalies in the pallidum, caudate, putamen, amygdala, and anterior cingulate [33]. The cortico-striatal-pallidum-thalamic and limbic circuits are considered vital in the pathophysiology of depression [34]. Although the right pallidum did not directly influence the right amygdala in our model, its effect was mediated by the influence of depression, the MSQ score, and the MIDAS score on amygdala volume. The importance of the pallidum in these relationships highlights its potential as a therapeutic target for improving anxiety, depression, and quality of life in patients with MG. Moreover, this may facilitate a new approach to managing patients with MG, whereby physical symptoms as well as emotional and psychological symptoms are considered linked to MG severity. Self-management strategies, such as relaxation techniques, may also be key to regulating anxiety and depression [35].

## Limitations

Our study has several limitations. The sample size of MG patients was limited. Studying a larger number of patients in future research will enable a more comprehensive understanding of MGs. Specifically, comparing MG patients with and without aura could elucidate specific MG mechanisms. Because our study was observational, we were unable to deduce causality. Thus, longitudinal or interventional studies will be needed. For example, future studies could investigate whether effective MG treatments also affect amygdala volume. Finally, we did not observe any differences in brain volume, disease severity, quality of life, depression, or anxiety between MG patients with and without aura. Therefore, further examinations of the functional disparities between the two subtypes of MG are necessary to determine the mechanisms underlying aura.

## Supporting information

**S1 File. Brain regions compared between MG patients and healthy subjects.**
(PDF)

**S2 File. Partial correlation analysis between variables in MG patients and healthy subjects.**
(PDF)

**S3 File. Table Direct, total and indirect effects of path analysis.**
(PDF)

## Acknowledgments

We thank Sarina Iwabuchi, PhD, from Edanz (https://jp.edanz.com/ac) for editing a draft of this manuscript.

## Author Contributions

**Conceptualization:** Shota Kosuge, Yuri Masaoka.

**Data curation:** Shota Kosuge, Yuri Masaoka, Hideyo Kasai, Nobuyuki Yoshii, Keiko Watanabe, Takaaki Naito, Hidetomo Murakami, Masahiko Izumizaki.

**Formal analysis:** Shota Kosuge, Yuri Masaoka, Motoyasu Honma, Kouzou Murakami, Miku Kosuge.

**Funding acquisition:** Yuri Masaoka.

**Investigation:** Shota Kosuge, Yuri Masaoka, Hideyo Kasai, Motoyasu Honma, Kouzou Murakami, Nobuyuki Yoshii, Keiko Watanabe, Takaaki Naito, Miku Kosuge, Misa Matsui, Daiki Shoji, Syunsuke Sakakura, Hidetomo Murakami, Masahiko Izumizaki.

**Methodology:** Yuri Masaoka, Kouzou Murakami, Takaaki Naito.

**Project administration:** Yuri Masaoka.

**Resources:** Yuri Masaoka.

**Software:** Yuri Masaoka.

**Supervision:** Yuri Masaoka.

**Validation:** Yuri Masaoka.

**Visualization:** Yuri Masaoka, Motoyasu Honma.

**Writing – original draft:** Shota Kosuge, Yuri Masaoka.

**Writing – review & editing:** Yuri Masaoka, Hidetomo Murakami, Masahiko Izumizaki.

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
