## [Decision Letter · Decision Letter 0]

7 Feb 2024

PONE-D-23-42993The Right Amygdala and Migraine: Analyzing Volume Reduction and Its Relationship with Symptom SeverityPLOS ONE

Dear Dr. MASAOKA,

Thank you for submitting your manuscript to PLOS ONE. After careful consideration, we feel that it has merit but does not fully meet PLOS ONE’s publication criteria as it currently stands. Therefore, we invite you to submit a revised version of the manuscript that addresses the points raised during the review process.

We look forward to receiving your revised manuscript.

Kind regards,

Keisuke Suzuki, MD, PhD

Academic Editor

PLOS ONE

Journal Requirements:

   "This study was supported by a JSPS KAKENHI Grant (Number 20K03482)"

  "JSPS KAKENHI Grant (Number 20K03482)"

   "JSPS KAKENHI Grant (Number 20K03482)"

Additional Editor Comments:

The authors are encouraged to revise the manuscript in accordance with the reviewers' comments.

Reviewers' comments:

Reviewer's Responses to Questions

**Comments to the Author**

1. Is the manuscript technically sound, and do the data support the conclusions?

Reviewer #1: No

Reviewer #2: Yes

2. Has the statistical analysis been performed appropriately and rigorously? 

Reviewer #1: I Don't Know

Reviewer #2: No

3. Have the authors made all data underlying the findings in their manuscript fully available?

Reviewer #1: Yes

Reviewer #2: Yes

4. Is the manuscript presented in an intelligible fashion and written in standard English?

Reviewer #1: No

Reviewer #2: No

5. Review Comments to the Author

Reviewer #1: I could not quite follow the meaning of "MIDAS scores influenced by MSQ outcomes which were further affected by depression, anxiety etc."

I thought it may be better for the authors to focus on the relationship between MRI finding and each questionnaire/assessments rather than the relationship between the questionnaire/assessments.

Please review the manuscript carefully as there seemed to be typos or incomplete sentence.

For example, I assume you do not need 5) in page 4, lint 74.

Reviewer #2: Overview and general recommendation:

Numerous neuroimaging studies have indicated gray matter volume reductions in migraine patients. And chronic migraine patients have different biomarkers compared to episodic migraine. The present study aimed to elucidate the association between gray matter volume changes and factors. The study hypothesized that there was a relationship between (1) gray matter volume (GMV) and migraine symptom severity and (2) GMV and psychological status. To test hypotheses, authors performed functional magnetic resonance imaging (fMRI) on patients and healthy controls. Group comparisons for intracranial volume were employed to validate structural differences between migraine sufferers and healthy controls. Then, path analyses were performed to study the interactions between variables. The findings suggest that the chronicity and severity of migraine headaches might specifically affect the right amygdala. The study highlights a complex interrelation that migraine disability is influenced by the quality of life, which is, in turn, affected by psychological states.

On the one hand, the present study contributes to neuroimaging and clinical research on migraine. The authors exhibit proficiency in data processing and analysis, and their findings extend the relationship between clinical symptoms and psychological status. However, upon closer examination, some areas for improvement in the paper were identified. Certain study aspects were described in insufficient detail, and some crucial points must be addressed.

Major concern:

1. Language. The author is requested to revise the entire text to make it consistent. (1) In the abstract, the author describes the use of 140 brain regions as the region of interest for analysis, but in the text, the author describes the use of 137 regions (page 6, line 111); (2) Add original word of the abbreviation "MG" (page 3, line 45); (3) Authors need to pay attention to whether the fifth point in the inclusion and exclusion criteria is written (page 4, line 74).

2. Quality control. It is recommended that the authors use tools to detect MRI image quality. The authors must explain the necessity of manually adjusting the segmentation [1]. If there is no statistical difference, please use the automated tool to ensure reproducibility; if there is a difference, it is recommended to perform a quality control evaluation in the supplementary material [2].

3. Statistical Analysis. (1) The author used 137 regions of interest. Did he use multiple comparison corrections in the analysis? Please add relevant content, and if not used, please explain. (2) Did the author test that the data is normal distribution? For example, using the Shapiro–Wilk test, (3) The migraine subjects recruited by the author were divided into migraine with aura (n = 11) and migraine without aura (n = 22). In the covariance analysis, is there " migraine with aura” a covariate? (4) Did the author conduct a collinearity test to ensure all variables are independent?

4. Interpretability. In the path analysis, the correlation between MIDAS and right amygdala was r=-0.38, a low correlation; the correlation between CESD and right pallidum volume was r=-0.35, which was also a low correlation. Please explain in the discussion: what is the difference between low and high correlation here? Does the low correlation indicate that the right amygdala volume is not closely related to MIDAS?

References:

[1] Vahermaa V, Aydogan DB, Raij T, et al. FreeSurfer 7 quality control: Key problem areas and importance of manual corrections. Neuroimage. 2023;279:120306. doi:10.1016/j.neuroimage.2023.120306

[2] Klapwijk ET, van de Kamp F, van der Meulen M, Peters S, Wierenga LM. Qoala-T: A supervised-learning tool for quality control of FreeSurfer segmented MRI data. Neuroimage. 2019;189:116-129. doi:10.1016/j.neuroimage.2019.01.014

6. PLOS authors have the option to publish the peer review history of their article (what does this mean?). If published, this will include your full peer review and any attached files.

Reviewer #1: No

Reviewer #2: No

---

## [Author Response · Author response to Decision Letter 0]

28 Feb 2024

Thank you for your useful comments and suggestion.

We corrected and added the manuscript according to your comments.

Reviewer #1

Question 1: I could not quite follow the meaning of "MIDAS scores influenced by MSQ outcomes which were further affected by depression, anxiety etc." I thought it may be better for the authors to focus on the relationship between MRI finding and each questionnaire/assessments rather than the relationship between the questionnaire/assessments.

Answer: Thank you for your suggestions. We modified the relationship between MIDAS, MSQ and other exogenous variables in the abstract section and a section created “Assumption from the path model” in the discussion. Since each path between MIDAS and MSQ, and MSQ and CED-D were significantly impacted, we would like to comment in the form of a proposal.

We made modifications as follow:

P2, L38: Abstract

Our path model suggests a complex relationship whereby migraine disability is strongly influenced by quality of life, which is, in turn, affected by psychological states, such as anxiety and depression.

P16, L272~

Assumptions from the path model

The path analysis also revealed several mediating factors that connected MG severity to lower amygdala volume. Our findings suggested that the MIDAS score, which represents MG severity, is shaped by patients’ quality of life, as measured by the MSQ. The MSQ is, in turn, influenced by exogenous factors, such as depression, anxiety, and the right pallidum volume. 

The association between the right pallidum volume and depression in MG patients was an intriguing result. This suggested that those with elevated depression levels tended to have lower pallidum volumes. The pallidum is a structure within the basal ganglia that plays a pivotal role in motor control and cognitive and emotional functions [32]. Depressed patients have been shown to exhibit structural and functional anomalies in the pallidum, caudate, putamen, amygdala, and anterior cingulate [33]. The cortico-striatal-pallidum-thalamic and limbic circuits are considered vital in the pathophysiology of depression [34]. Although the right pallidum did not directly influence the right amygdala in our model, its effect was mediated by the influence of depression, the MSQ score, and the MIDAS score on amygdala volume. The importance of the pallidum in these relationships highlights its potential as a therapeutic target for improving anxiety, depression, and quality of life in patients with MG. Moreover, this may facilitate a new approach to managing patients with MG, whereby physical symptoms as well as emotional and psychological symptoms are considered linked to MG severity. Self-management strategies, such as relaxation techniques, may also be key to regulating anxiety and depression [35].

Question 2: Please review the manuscript carefully as there seemed to be typos or incomplete sentence. For example, I assume you do not need 5) in page 4, lint 74.

Answer: We corrected typos and incomplete sentences throughout the manuscript.

Thank you for your useful comments and suggestion.

We corrected and added the manuscript according to your comments.

Reviewer #2

Major concern:

Question 1: Language. The author is requested to revise the entire text to make it consistent. (1) In the abstract, the author describes the use of 140 brain regions as the region of interest for analysis, but in the text, the author describes the use of 137 regions (page 6, line 111).

Answer: The manuscript revised the entire text, and we all corrected to 140 brain regions throughout the manuscript.

Questions 2: Add original word of the abbreviation "MG" (page 3, line 45)

Answer: We added the abbreviation “ MG” as follow.

P3, L2~ Introduction

Migraine (MG), a prevalent neurological disorder marked by severe headaches and an array of neurological symptoms, poses a significant public health challenge worldwide [1,2].

Question 3: Authors need to pay attention to whether the fifth point in the inclusion and exclusion criteria is written (page 4, line 74).

 Answer: We corrected as follow: 

P4, L68-73.

Thirty-three patients with MG participated in the study. Eligibility criteria were as follows: 1) aged 21 to 60 years, 2) experienced MG attacks for a minimum of 2 years, 3) no clear structural pathology or hyperintense white-matter lesions on previous magnetic resonance imaging (MRI) scans, 4) no comorbid basal ganglia disorder or history of other neurological disorders.

Question 4: Quality control. It is recommended that the authors use tools to detect MRI image quality. The authors must explain the necessity of manually adjusting the segmentation [1]. If there is no statistical difference, please use the automated tool to ensure reproducibility; if there is a difference, it is recommended to perform a quality control evaluation in the supplementary material [2].

Answer: Thank you for your suggestion. We also had the automated freesurfer data so we compared pre- and post-manual quality control. We added following statistic results, explanations and a reference you suggested in the method. Also mean and standard deviation from each brain regions extracted from original Freesufer data were indicated in the Supplemental information.

P6, L104~

Manual corrections were confined to eliminating non-brain tissue that appeared within the cortical boundary, according to our written guidelines established to ensure consistency. Most errors were located in the paracentral lobule and near the cerebellar tentorium. After any necessary adjustments were made, the images were reprocessed using the recon-all command. According to quality control studies of structural images [26], differences between pre- and post-quality control measurements are small. Nevertheless, we ensured this by comparing all 140 gray matter volumes from the stats files using a paired t-test and a reliablility analysis between pre- and post-manual quality control. There were no differences in gray matter volumes between pre- and post-manual quality control (all P > 0.05) across all participants. We confimed that Cronbach’s alpha was all >0.9 (all mean and standard deviation were indicated in Supplementary Table 1-5). We used the automatically extracted volume to ensure reproducibility.

[26] Vahermaa V, Aydogan DB, Raij T, et al. FreeSurfer 7 quality control: Key problem areas and importance of manual corrections. Neuroimage. 2023;279:120306. doi:10.1016/j.neuroimage.2023.120306

Question 5: Statistical Analysis. (1) The author used 137 regions of interest. Did he use multiple comparison corrections in the analysis? Please add relevant content, and if not used, please explain. 

Answer: We added sentences as follow:

P8.L131~

The main objective here was to examine the relationship between symptom severity and the neural substrates underlying symptom severity. Thus, we constrained our hypotheses according to the anatomical connectivity relevant to the MIDAS. As such, gray matter volume differences between MG patients and controls were analyzed using ANCOVA without multiple comparison correction (e.g., family-wise error correction) to identify brain regions relevant to MG.

Question 6: Did the author test that the data is normal distribution? For example, using the Shapiro–Wilk test

Answer: We performed Shapiro–Wilk test for MIDAS, MSQ, MoCA-J scores, CESD and STAI scores to confirm normality of distribution. For ANCOVA, equality of hometogeneity of variance was confirmed by Levene’s test.

We added sentences as follow:

P7, L123~

All data were first analyzed using the Shapiro–Wilk test to test the normality of the distribution. Differences in age, years of education, and MIDAS, MSQ, and MoCA-J scores (Shapiro–Wilk, P < 0.05) were analyzed using the non-parametric Mann–Whitney test, and CESD and STAI scores (Shapiro–Wilk, P > 0.05) were analyzed using parametric t-tests.

P11, L180~

The homogeneity of variance was equal across groups as confirmed by Levene’s test (all P > 0.05) in the ANCOVA, which indicated that parametric analysis was suitable.

Question 7: The migraine subjects recruited by the author were divided into migraine with aura (n = 11) and migraine without aura (n = 22). In the covariance analysis, is there " migraine with aura” a covariate?

Answer: We added following sentences in the statistical analysis section.

P7, L128~

Because there was no difference in volume between patients with MG with or without aura (as noted in the results section), the presence of aura was not included as a covariate for the ANCOVA.

Question 8: Did the author conduct a collinearity test to ensure all variables are independent?

Answer: We added following senteces indicated collinearity test results.

P12, L195~

For the multiple regression analysis, we examined multicollinearity for each independent variable to ensure the variables were not correlated. The variance inflation factor was < 5 for all independent variables (left pallidum: 3.2, right pallidum: 2.9, and bilateral amygdala: 1.6). 

Question 9: Interpretability. In the path analysis, the correlation between MIDAS and right amygdala was r=-0.38, a low correlation; the correlation between CESD and right pallidum volume was r=-0.35, which was also a low correlation. Please explain in the discussion: what is the difference between low and high correlation here? Does the low correlation indicate that the right amygdala volume is not closely related to MIDAS?

Answer: Thank you for your questions. It is very important to note about each coefficient but we did not indicate details. 

Straight arrows indicate the direction of causal relationships between variables, and indicate standardized beta coefficients. On the other hand, double-headed curved arrows indicate correlations between exogenous variables, indicated r. 

We added following descriptions in the statistical analysis, results sections and figure legends.

Statistical Analysis, P8, L148~

The structural equation modeling software IBM SPSS Statistics Amos (version 23.0, IBM Corp.) determined the statistical significance of the path coefficients. Straight arrows linking the variables indicate the direction of causal relationships between variables. Double-headed curved arrows indicate correlations between exogenous variables. Standardized beta coefficients are indicated by straight arrows.

Results, P13, L223~

In the path model (Fig 2), the three exogenous variables (i.e., state anxiety score, CESD score, and right pallidum volume) were modeled as being correlated, as depicted by the double-headed arrows (indicated r as a correlation coefficient). The endogenous dependent or independent variables were connected by one single-headed arrow (indicated as betas [standardized regression weights]). There was a covariate relationship between the CESD score and state anxiety score (r = 0.81, P < 0.001) and the right pallidum volume (r = −0.35, P = 0.01). The CESD score had a direct effect on the MSQ score (β = 0.58, P < 0.001). The MSQ score had an impact on the MIDAS score (β = 0.71, P < 0.001), and the MIDAS score had an impact on the right amygdala (β = −0.38, P = 0.002). This suggested that a higher MIDAS score corresponded with a smaller right amygdala volume.

Figure legends, P14, L230~

Fig 2. Path analysis with standardized regression weights (β; single-headed arrows) and correlation coefficients (double-headed arrows) for the migraine (MG) group. The Migraine Disability Assessment (MIDAS) score followed a standardized direct path to the right amygdala, which suggested that a higher MIDAS score had an impact on the right amygdala volume. The MIDAS score was directly influenced by the Migraine Specific Quality of Life Questionnaire (MSQ) score, and the MSQ score, in turn, had a direct path originating from the Epidemiologic Studies Depression Scale (CESD) score. The CESD score was correlated with the state anxiety score and the right pallidum volume.

---

## [Decision Letter · Decision Letter 1]

18 Mar 2024

The Right Amygdala and Migraine: Analyzing Volume Reduction and Its Relationship with Symptom Severity

PONE-D-23-42993R1

Dear Dr. MASAOKA,

We’re pleased to inform you that your manuscript has been judged scientifically suitable for publication and will be formally accepted for publication once it meets all outstanding technical requirements.

Kind regards,

Keisuke Suzuki, MD, PhD

Academic Editor

PLOS ONE

Additional Editor Comments (optional):

Reviewers' comments:

Reviewer's Responses to Questions

**Comments to the Author**

1. If the authors have adequately addressed your comments raised in a previous round of review and you feel that this manuscript is now acceptable for publication, you may indicate that here to bypass the “Comments to the Author” section, enter your conflict of interest statement in the “Confidential to Editor” section, and submit your "Accept" recommendation.

Reviewer #1: All comments have been addressed

2. Is the manuscript technically sound, and do the data support the conclusions?

Reviewer #1: Yes

3. Has the statistical analysis been performed appropriately and rigorously? 

Reviewer #1: I Don't Know

4. Have the authors made all data underlying the findings in their manuscript fully available?

Reviewer #1: Yes

5. Is the manuscript presented in an intelligible fashion and written in standard English?

Reviewer #1: Yes

6. Review Comments to the Author

Reviewer #1: Authors addressed to the commnets accordingly.

Thank you very much for your hard work upon revising the manuscript.

7. PLOS authors have the option to publish the peer review history of their article (what does this mean?). If published, this will include your full peer review and any attached files.

Reviewer #1: No

---

## [Editor Report · Acceptance letter]

24 Mar 2024

PONE-D-23-42993R1 

PLOS ONE

Dear Dr. Masaoka, 

I'm pleased to inform you that your manuscript has been deemed suitable for publication in PLOS ONE. Congratulations! Your manuscript is now being handed over to our production team.

Kind regards, 

on behalf of

Dr. Keisuke Suzuki 

Academic Editor

PLOS ONE